# Comparison of the Efficacy and Welfare of Different Training Methods in Stopping Chasing Behavior in Dogs

**DOI:** 10.3390/ani14182632

**Published:** 2024-09-11

**Authors:** Anamarie C. Johnson, Clive D. L. Wynne

**Affiliations:** Department of Psychology, Arizona State University, P.O. Box 871104, Tempe, AZ 85287, USA; cwynne1@asu.edu

**Keywords:** dog training, aversive training, behavior concerns, animal welfare

## Abstract

**Simple Summary:**

The use of electronic shock collars (“e-collars”) is one of the most controversial topics in dog training. In this study, we compared e-collars to methods relying entirely on food rewards in order to stop dogs running after a lure. We found that dogs receiving shocks from e-collars stopped chasing a lure within two sessions of ten minutes of lure running per session. These dogs also refrained from chasing the lure in three out of four test sessions. Two groups of dogs trained with food reward did not refrain from chasing the lure across five training sessions and failed all four test sessions. Aside from presumably pain-induced yelps in the dogs with e-collars when they received shocks, none of the dogs in any groups showed any signs of stress or distress. E-collars may be an appropriate tool in the hands of expert trainers training behaviors that have important welfare impacts, such as running after cars or other animals. Future studies should investigate the levels of expertise needed to use e-collars effectively, the kinds of behavioral problems for which they are best suited, and the longer-term implications of their use.

**Abstract:**

Controversy surrounds the efficacy and welfare implications of different forms of dog training with several studies asserting that electronic shock collars have negative welfare impacts while not being more effective than non-aversive methods. However, these studies did not specify the schedule and intensity of punishment used or the effectiveness of the training method. In the current study, we attempted to train dogs across six sessions to desist from chasing a fast-moving lure in one of three randomly assigned conditions and then tested for retention and generalization in four further test trials. Group A was trained with e-collars; Group B was trained with non-aversive methods and the lure moving as with Group A; and Group C was trained as for Group B but with the lure initially moving slowly before its speed was progressively increased. All dogs in Group A stopped running towards the lure after one or two sessions, and none chased the lure in the first three tests: 67% of these dogs chased the lure in the final test in a novel arena. None of the dogs in the either Group B or C successfully refrained from chasing the moving lure in training or any test. Video behavior coding indicated few stress-related behaviors across the training groups.

## 1. Introduction

Since the 1930s, dog handlers have been using electricity to punish dogs for problematic behavior. A device patented in 1935 administered shock down an electrified leash that was connected to an electric generator; the inventor noted that the length of the leash allowed for a handler to administer a punishment from a distance [1]. A 1955 patent noted its improvement in previous electric training devices by not requiring the handler to be connected to the dog by physical attachments and claimed it was more humane than previous models which had been criticized for their apparent cruelty [2]. The handler held a radio transmitter, and the dog wore a harness containing the radio receiver. Once a signal was relayed, it activated a series of coils and capacitors to generate a voltage potential to shock the dog via electrodes built into the harness. By 1961, the electric aspects of the collar were small enough that batteries could be stored in the side of the collar to administer a shock to a dog [3]. Over the next decades, these collars continued to be updated and modified to the range of brands and types available today.

As dog handlers and their dogs were beginning to utilize electricity in the field, the administration of electric shocks to impact behavior was also being investigated in the laboratory. Solomon and Wynne placed dogs in a box with an electrified floor [4]. Shocks through the floor were preceded by the offset of a light in the box, and the dogs could avoid the shock by jumping over a barrier into a distinct section of the box (labeled “traumatic avoidance learning”). Dogs were shocked at the highest possible level without causing their legs to involuntarily contract. Some of the observed behavioral reactions to the shock included high-pitched vocalizations, urination, defecation, and salivation. The investigators’ focus was on the change from escape behavior (jumping the barrier once the shock occurred) to avoidance behavior (jumping on the light offset, in advance of the shock) [4].

More recently, Christiansen and colleagues studied the impact of e-collars in reducing problematic behavior in dogs [5]. The chasing motivation of 114 pet dogs (all herding breeds) was assessed by exposing them to free-roaming sheep. Each dog was fitted with an electronic collar and a long, dragging leash for the sheep’s safety in case shock administration did not stop chasing behavior. Dogs received shocks if they were between one and two meters of a sheep, and shocks were repeated if the dogs did not leave that proximity zone or re-approached the sheep. Only 14 of the 114 dogs approached the sheep within the criterion distance and, thus, received shocks. The dogs were then re-exposed to sheep a year later to observe any learning or other behavioral effects. Dogs that had received shocks in the first year showed a larger reduction in their probability of chasing sheep and observed the sheep faster in the second year, thus demonstrating a learning effect [5]. Owners reported that, in the intervening year, dogs that had received electric shocks in the study did not demonstrate any poorer welfare or behavioral changes compared to dogs that had not received shocks. Interestingly, mere exposure to the sheep appeared to reduce interest as both dogs that did and did not receive shocks in the first year had less interest in sheep in year 2 [5].

Another study aimed to observe the immediate behavioral responses and potential long-term impacts on dogs receiving shocks during training [6]. Of the 46 police or working dogs recruited, 32 received a total of 107 shocks and were observed lowering their ears, yelping, licking their lips, and, for a few dogs, snapping at the handler during different exercises [6]. A total of 15 control and 16 shocked dogs were then observed to compare if there were any behavioral differences between the groups. Overall, the authors noted that the shocked dogs displayed more stress-related behaviors than the control dogs at the training ground and that the shocked dogs were also more stressed than the control dogs at a park with no association with training, suggesting that their overall welfare was impaired [6]. They also concluded that given the observed behavior, when the dogs were experiencing shock, they were experiencing pain, not just reacting to an irritant [6].

In a laboratory study assessing the intensity of physiological stress responses to e-collar stimulation, Schalke et al. exposed three groups of laboratory beagles to electric shocks [7]. In the “aversion group”, dogs were shocked for moving towards a simulated prey object; dogs in the “here group” were shocked if they moved towards the prey object after receiving a “here” command; finally, in a “random group”, dogs were shocked arbitrarily in the presence of the prey stimulus. All the dogs received shocks at the highest arbitrary level of the collar to mimic the worst possible welfare scenario. Investigators measured both heart rate and salivary cortisol to determine welfare impact. The dogs in the random group that were unable to predict when electric shock stimulation might occur had the highest increase in cortisol levels of the three groups. The dogs in the other two groups showed smaller increases in cortisol levels; those in the aversion group showed the smallest cortisol increase, presumably because their shocks were paired with their approach toward the prey stimulus, so they could control their behavior in relation to it. Dogs in the here group were initially trained for recall behavior without any prey present, so while they could predict the administration of the stimulus, they did not associate it with chasing prey [7].

In another study with 42 police dogs, Salgirli et al. investigated the behavioral and physiological effects of three training methods on stopping a behavior [8]. In a within-subject design, researchers assessed the effectiveness of an e-collar, a pinch collar, and a conditioned quitting signal. The nature of the quitting signal was not described, but it was designed to evoke frustration by signaling the withdrawal of a reward. To proceed in the study, the dog had to withdraw from a toy immediately following onset of the signal. The dogs were corrected with positive punishment (the addition of an aversive stimulus to reduce the likelihood of a behavior recurring) if they broke a “heel” command and went towards a decoy handler. The dogs were tested an hour after training to assess whether the administration of the training method was successful in the dog maintaining the heel. All the sessions were recorded for behavioral analysis, and salivary cortisol samples were taken at different time points [8]. The e-collar had the best learning effect for 39 of the 42 dogs, while the pinch collar had a learning effect for 32 out of the 42 dogs. The quitting signal was only tested in four dogs and was only successful in three since the remaining 38 dogs never responded to it [8]. Vocalizations were most frequent in the e-collar condition compared to the pinch collar condition, but more dogs had backward-pointing ears in the pinch collar condition than in the e-collar condition (though this latter difference was not statistically significant). There were no statistically significant differences in salivary cortisol levels across the three conditions [8].

A study in New Zealand investigated the effectiveness of aversion training that is required in that country before dogs are allowed into kiwi habitats [9]. One hundred and twenty hunting dogs were recruited to investigate the efficacy of kiwi aversion training (KAT) over various time points. In KAT, dogs are exposed to frozen or stuffed whole kiwi and kiwi feathers and receive shock stimulation when they contact the stimuli. If they attempt to contact the stimuli a second time, they were shocked again. A total of 65 dogs were tested for aversion to the stimuli immediately after training, 15 from that cohort were also tested a month later, and a different set of 55 dogs were tested one year after KAT training. The dogs demonstrated significantly higher interest in the stimuli prior to training than immediately after. The dogs tested a month after training showed a strong-to-moderate aversion to the stimuli. Only 12.7% of the dogs tested one year after their training showed any interest in the kiwi stimuli, and after receiving a shock in response to that interest, they demonstrated strong aversion to the stimuli [9]. A follow-up study analyzed the KAT training records for nearly 1200 dogs [10]. In the first training session, 89% of the dogs needed only one shock to avoid the stimuli. There was no significant difference in aversion behaviors between the first and second year after training, but there was less aversion three years later [10].

A group at the University of Lincoln in the United Kingdom published two reports from a study comparing e-collar training and positive reinforcement over five days in dogs with recall and livestock chasing concerns [11,12]. Sixty-three dogs were divided into three groups. All groups received positive reinforcement of either food, praise, or play at the trainer’s discretion [12]. A total of 21 dogs received e-collar training or leash pressure from practicing e-collar trainers if they did not appropriately respond to a cue (Group A), 21 received either positive-reinforcement training with rewards or negative reinforcement with leash pressure from the same e-collar trainers as Group A (Group B), and the remaining 21 received solely positive-reinforcement training from practicing positive-reinforcement trainers (Group C) [12]. The dogs were motivated to chase or move with distractors like penned livestock or other on-leash dogs [11].

Both salivary and urinary cortisol samples were taken, and while there was a group effect on salivary cortisol, there were no significant differences in urinary cortisol. Group C (positive reinforcement only) dogs had higher salivary cortisol levels during the study than those in Groups A and B [12]. Dogs in Group A (e-collar) yawned more than those in Group C and demonstrated more sudden head and body movements away from the trainer [12]. Yelping was more common in Group A but occurred relatively infrequently. Group C dogs responded faster to the “come” and “sit” cues compared to the other groups and required fewer cues to illicit a response [11]. The authors of both papers concluded that positive-reinforcement training was as effective as aversive training [11] and resulted in less distress [12].

Sargisson and McLean [13] noted several problems with the China et al. [11] and Cooper et al. [12] reports. First, they noted that dogs in the positive-reinforcement group (Group C) were trained at a different time, a different location, and by different trainers than the other groups. Second, they observed that previous research recommended that punishment should be delivered with a maximum acceptable intensity level which was not the case in China et al. [11] and Cooper et al. [12]. The critique also noted that the reported problematic behavior which led to dogs being entered into the study included aggression to other dogs, whereas the training program concerned obedience, and the dogs were already obedient even in the first training sessions. Furthermore, Sargisson and McLean noted the absence of any baseline assessment of the dogs’ behavior, making it hard to judge the efficacy of the training program [13]. Overall, while Sargisson and McLean acknowledged that the use of intense shock is ethically concerning, if a dog that is at risk for euthanasia for problematic behavior can be stopped with a few appropriately delivered shocks, then this course of action may be “morally appropriate” [13] (p. 2). Although Cooper et al. [12] and China et al. [11] concluded that their findings indicated that e-collar training posed a risk for suffering, they did not report any negative longer-term welfare consequences for the dogs in any group in their study.

Besides studies that have focused on e-collar use, several studies have examined more general impacts of aversive training methods and have reported that dogs experiencing aversive training may have less attachment to their owners [14], show a more pessimistic perspective [15,16,17], and have overall poorer welfare than dogs trained with strictly positive-reinforcement methods [18,19].

Notwithstanding questions of e-collar efficacy and welfare implications, they continue to be popular devices to modify problematic behavior. In a survey of dog owners in the United Kingdom where there have been attempts to ban these devices, 3% still reported using a remote e-collar [20]; in France, 26% of dog owners responded that they used these devices [21].

A dog’s motivation to chase or run is a concern for many dog owners due to the risk of absconding, injuring other animals, or being injured in traffic [22,23]. A leash is a simple way to prevent a dog from straying, but in an emergency, owners need a method to ensure that their dog will return to them. There are several ways that practicing trainers stop problematic chasing and absconding either using aversive methods through e-collar stimulation, pairing the aversive shock with the stimulus eliciting chasing [24,25], or non-aversive methods offering rewards for alternative behaviors like “sit” or “stay” [26,27,28]. One of the reported advantages of e-collar training compared to non-aversive methods is that it is fast and easy to implement from a distance [24], whereas non-aversive methods will take longer and require closer management of the dog.

The aim of the current study was to improve understanding of the efficacy and welfare impacts of e-collar training versus non-aversive methods for dogs prone to chase moving objects. In response to criticism of some prior studies, we used the same two trainers for all training conditions and, as far as practicable, designed protocols with single contingencies. We measured relative training success in precisely defined training protocols with and without e-collars, and we also tested generalization of training in four tests. Welfare was assessed through fecal cortisol sampling and behavioral analysis.

## 2. Materials and Methods

This study was approved by the Institutional Animal Care and Use Committee (IACUC) of Arizona State University (protocol number: 22-1976R).

### 2.1. Participants and Location

Thirty dogs were recruited for this study via an online survey distributed on social media. Questions ranged from general household information and how many dogs lived in the home to specific questions about each dog’s history (Appendix A). The aim of this survey was to recruit chasing-prone dogs with limited to no exposure to an e-collar, but with owners who were open to e-collar training. Only medium-to-large-sized dogs over the age of 6 months were invited to participate.

This study took place over six weeks between 19 February 2023 and 31 March 2023, at a training facility in Plant City, Florida. A training arena was set up on a large lawn at one section of the property (Figure 1). The arena measured 53 m by 23 m (approximately 175 feet × 75 feet) and was demarcated by plastic fencing to ensure that the dogs stayed in the training space. The trainer, the owner, and the dog positioned themselves at one end of the arena approximately 45 m away from the lure course located at the opposite end of the field.

### 2.2. Study Trainers

Two trainers worked with all the dogs across all the groups. The first trainer has over 30 years of experience with a focus on competitive dog sports and has coached in a variety of fields including service dog agencies, police departments, and private security. He has coached for and operates a certification course for new dog trainers focusing on learning theory and practical application of aversive and non-aversive methods. The second trainer has worked with dogs for over five years, received his certification through the first trainer’s program, and currently operates a dog training business focusing on service and pet dog training as well as competitive dog sports. Both trainers have experience with e-collar training and utilize an e-collar as one aspect of their overall training method.

### 2.3. Materials

We used the Garmin Sport Pro e-collar (Garmin International, Inc., Olathe, KS, USA). This device has stimulation levels ranging from 1 to 10 arbitrary units. The collar was fitted on each dog to be snug enough for sufficient contact with the dogs’ skin. The device was only active when the dogs were in the training arena.

The lure course (Pinelli Machine Works, Etowah, NC, USA) utilized power tool batteries to energize a motor which moved twine through two wheels on the controller box and round five pulley wheels set into the grass of the training area. A thin piece of colored plastic was tied onto the twine to create a visible moving lure for the dogs to chase. The lure could be run at a variety of speeds, labeled “low”, “medium”, and “high”. The lure could also reverse its direction.

### 2.4. Procedure

All dogs, regardless of group assignment, wore e-collars throughout the five days of study before, during, and after sessions to control for a possible confounding effect of the weight of the collar.

The dog, trainer, and dog’s owner were present throughout all sessions for all the groups.

Figure 2 shows the flow in which the recruited dogs moved through this study and the respective time points of training.

#### 2.4.1. Day 1

On Day 1, the dogs were tested for their motivation to chase the lure. Each dog was given up to twenty minutes to run around the training arena and engage with the moving lure. The dogs were then given at least a two-hour break to rest before another session where they chased the lure.

Only dogs that consistently chased the lure continued in this study and were randomly assigned to one of three study groups: Group A received e-collar stimulation and were exposed to the lure at high speed, Group B received food rewards and were exposed to the lure at high speed, and Group C received food rewards and were exposed to the lure at gradually increasing speeds.

#### 2.4.2. Training

##### Stopping Word

Every group was exposed to the word “Banana” as a signal to stop moving towards the lure. We chose “Banana” because it had likely no previous association for dogs in a training environment, unlike more common training words such as “stop”, “come”, or “here”.

##### Group A Procedure: Day 2–Day 4

Each session ran for a goal of 20 min, with a maximum of 10 min of lure exposure per session. The lure was run for 2 min, followed by a minimum 2 min intertrial interval (ITI). The sessions were separated by a minimum of two hours.

Day 2, Training Session 1 was the first day of training for dogs in the e-collar group. At the start of the session, the handheld e-collar remote was set at level 6 out of 10 arbitrary units. The dog, the trainer, and the dog’s owner entered the training arena at the starting point (Figure 1) and waited for the lure to be run at its highest speed.

When the dog ran towards and contacted the lure, the trainer said “banana” and deployed one shock to the dog; contact and proximity to the lure were important to classically condition the dogs to associate contact with the lure to receiving a shock as well as the pairing of the novel word “banana”. The lure continued moving unless it was disabled by the dog. If the dog continued to attempt to interact with the lure, the trainer repeated “banana” and administered another shock. The intensity level of the e-collar was either increased or decreased by the trainer based on his perception of the dog’s motivation, gauged by running speed and orientation to the lure. Shock administration was repeated until the dog stopped going after the lure and moved away back towards the starting point.

This procedure continued throughout the subsequent days and sessions. Once the dog stopped running at high speed towards the lure, if it approached the pre-determined threshold line (Figure 1), the trainer said “banana” but did not administer a shock. If the dog appropriately moved away from the lure, then no other feedback was provided. If the dog did not move away from the lure, then the trainer said “banana” again and administered a shock.

To ensure that the dog did not develop any conditioned fear to the training arena beyond the avoidance of the lure, between sessions, the trainer and owner walked around the field, as needed, and encouraged the dog to move around.

##### Group B and Group C Procedure: Day 2, Word Conditioning

The goal of the word conditioning session was to associate the word “banana” with the presentation of a high-value treat. The owner, trainer, and dog were in an enclosed x-pen area at the starting point of the course (Figure 1). The dog was off-leash and allowed to move around the penned area. The lure was off during this conditioning session, but the dogs were aware of its location within the training arena.

The session ran for twenty minutes and consisted of five two-minute active trials where a high-value treat was dropped in a bowl every five seconds and separated by a minimum two-minute ITI. During an active trial, the trainer said “banana”, then dropped a treat in a metal bowl.

##### Group B Procedure: Day 2, Session 2–Day 4

The second session of Day 2 was the first day of training with the lure course. The goal of this training was to utilize the conditioned association of “banana” and the presentation of a treat to reward the dog correctly moving away from the lure. The dog only received the food reward if it correctly responded to “banana” and went back to the start position away from the lure. As for the dogs in Group A, the lure in Session 2 was presented at its highest speed. If the dog ran towards the lure, the moment the dog crossed the pre-determined threshold line, the trainer said “banana” and dropped a treat in the metal bowl at the starting point 23 m away. If the dog continued to run towards the lure, the trainer continued to say “banana” every five seconds on average until the dog responded. If the dog did not return to receive the treats, the trainer picked up the treats from the bowl.

The lure ran for the full two-minute trial. The sessions ran for a maximum of five active two-minute trials. At the end of the two-minutes, there was a minimum two-minute ITI. This procedure continued throughout the subsequent days and sessions.

##### Group C Procedure, Day 2, Session 2–Day 4

As in Group B, the goal of this training was to utilize the conditioned association of “banana” and the presentation of treat to reinforce the dog moving away from the lure. The second session of Day 2 was the first day of training with the lure course, but for dogs in this group, the lure was stationary and did not increase to higher speeds unless the dog successfully moved away from it in four out of five responses in one session when off-leash in the training arena. The moment the dog crossed the pre-determined threshold (Figure 1), the trainer said “banana” and dropped a food reward in a bowl. The trainer continued to say “banana” every five seconds on average and dropped treats into the bowl. If the dog did not return to receive the treats, the trainer picked up the treats from the bowl.

As in Group A, owner and trainer walked around the field, as needed, and encouraged the dog to move around.

Once the dog successfully responded to “banana”, the lure was moved up to “medium” speed. “Low” speed was not used because dogs would catch or destroy the lure at the lowest speed. Similarly, for some dogs, the lure had to be increased to high speed prior to the dog successfully responding to “banana” to stop the dog from catching and destroying it.

#### 2.4.3. Testing

Day 5 was the test day. All dogs experienced the same four tests: three of which took place in the training arena and one in a novel area at a different section of the training facility. The dogs did not receive any punishment or reinforcement during the tests.

##### Test 1

Test 1 took place in the training arena and most resembled the training sessions. The dog, owner, and trainer were located at the starting point, and the lure was deployed. The moment the dog crossed the pre-determined threshold, the trainer said “banana”. If the dog appropriately responded to the word or did not cross the threshold line, then the test was successful. If the dog continued to run and chase the lure, then the test was unsuccessful.

##### Test 2

Test 2 took place in the training arena. In this test, the dog, owner, and trainer were again located at the starting point and the lure was deployed. This time if the dog crossed the threshold line, “banana” was not said. Scoring was the same as for Test 1.

##### Test 3

Test 3 was the final test in the training arena. The owner and trainer led the dog to the start location and then released the dog. Then, the owner and trainer stepped away, and the dog was alone in the training arena. The scoring was the same as for Test 1.

##### Test 4

Test 4 took place in a novel area and with a novel lure, a soft toy attached to the lure twine. Since the toy would not slide around the pulleys, this lure course was a straight line that ran the width of the course at the end opposite the starting point. The novel arena had the same dimensions as the training arena. Test 4 was otherwise identical to Test 1 and was scored the same way.

### 2.5. Cortisol Collection

Fecal samples were opportunistically collected for each dog on each day. About 15 mL of feces was collected from a fresh sample and placed in a Ziploc bag (SC Johnson, Racine, WI, USA) and then immediately frozen to −17.8 °C. All samples were shipped on dry ice to a testing laboratory for a cortisol assay [29].

### 2.6. Video Analysis and Ethogram

All the sessions were recorded using Akaso EK7000 cameras (Akaso, Frederick, MD, USA). Three cameras were set up around the training arena to capture dog behavior from different angles (Figure 1). A fourth camera was mounted on the e-collar remote to record when the shocks were administered and at what intensity. A single camera was used to record the conditioning sessions of Groups B and C.

The videos were analyzed using the event logging software BORIS (Version 8.27, Torino, Italy) [30], operated by video coders trained to 80% or better accuracy. The video coders were blind to the aims of this study. Twenty percent of the videos were double-coded to ensure reliability. Behaviors were coded as either “state” where the duration of each observed behavior was recorded or “point” where the behavior was coded for the number of times it was observed (Table 1).

### 2.7. Data and Statistical Analysis

State behaviors were calculated for each dog as a proportion of time spent in the behavior during the active trials of a session. Behaviors were not coded or analyzed during intertrial intervals. Point behaviors were calculated for each dog as a frequency during active trials to the equivalent rate per hour.

All statistical analyses were conducted using the software package SPSS (Version 29, IBM, Chicago, IL, USA).

Due to the small sample size and missing data points, a non-parametric Kruskal–Wallis test was conducted to compare cortisol levels across the three groups. Kruskal–Wallis tests were also used for testing differences in vocalization behaviors (barking, yelping, and whining) between groups as these were also not normally distributed.

Mixed-method ANOVAs were conducted to compare the percentage of time engaged in the most commonly occurring state behaviors across the groups and across the different training sessions. Pairwise t-test comparisons with Bonferroni corrections were completed to locate any significant differences between groups, and simple effects testing was conducted to examine any significant interaction effects.

## 3. Results

### 3.1. Participants

Thirty dogs were recruited and invited to the study site. Of these, only 19 dogs were consistently motivated to chase and completed the remaining study days. Eight dogs were assigned to Group A, five to Group B, and six to Group C. Two dogs were removed from Group A as they received e-collar stimulation in excess of the 20 shocks approved by the IACUC, leaving six dogs in this group. The inclusion of these dogs’ data would not have altered the overall conclusions of this study.

Ages ranged from 8 months to 5 years old. All the dogs had zero to very limited experience with e-collar stimulation (at most, some owners may have introduced a collar once or twice to the dog but did not continue with e-collar training) (Table 2).

Most of the shocks were administered during the first session of training (Figure 3). The frequency steeply declined by Session 2, and very few shocks were administered over the remaining training days. The intensity level varied over the training (range 3–10), with an average level across all dogs of 5.68. Efficacy in e-collar training was also observed in the decreased rate of running which is discussed in more detail in the Section 3.4.

There was no significant difference in the rate per hour of eating an offered treat, between Groups B (M = 420.89, SD = 281.43) and C (M = 469.59, SD = 167.87) (independent samples *t*-test, t (8) = −3.47, *p* = 0.28).

Group B dogs never successfully responded to “banana” by stopping their chasing, as evident in percentage of time spent running (Section 3.5). Three of the six dogs in Group C responded to “banana” by returning to the starting point when the lure was stationary. However, once the lure began moving, these dogs ignored “banana” and chased it. Two other dogs in Group C showed mixed responses to the stationary lure—sometimes, they stopped moving towards the lure on hearing “banana” but did not return for the treat; at other times, they stopped twice in succession but then did not respond on subsequent commands—thereby never meeting the criterion of four out of five successful responses in a single session. The final dog in this group either laid on top of the motionless lure and did not respond to “banana” or laid at the starting point and did not follow the owner to approach the threshold line. For these three dogs, in the final training sessions, the lure moved and, once it was moving, they did not respond to “banana”.

### 3.2. Test Performance

Over the four tests, the dogs trained with the e-collar outperformed the dogs trained with food rewards. For Tests 1–3 that occurred in the training arena, e-collar-trained dogs were 100% successful; however, success varied on Test 4 which was conducted in a novel area with a novel lure. Of the four e-collar-trained dogs that were successful in Test 4, two successfully stopped chasing the novel lure upon hearing “banana”, and two did not attempt to chase it. The dogs in Group B and C were not successful in any of the tests.

### 3.3. Cortisol Analysis

Only 9 of the 19 dogs provided a fecal sample on each study day: four from Group A, two of which were removed from analysis because of the exceeded shock levels; one from Group B; and four from Group C.

Since peak fecal cortisol is observed 24 h after stimulation [31], the levels obtained on a given day reflect cortisol levels from the day prior. Consequently, the samples collected on Day 1 reflect the dog’s stress levels prior to arriving at the training site, the samples collected on Day 2 reflect the cortisol levels from Day 1, and so on. Figure 4 shows the mean fecal cortisol levels of each group for each day of training. There were no significant differences in cortisol levels across the three groups (Kruskal–Wallis H(2) = 1.87, *p* = 0.39).

### 3.4. Behavior Analysis

Statistical analysis of the six most frequently occurring state behaviors—running, walking, staring and orienting to the lure, being away from the lure (start), stalking, and engaging in alternative behaviors—examined the main effect differences in groups, training sessions, and their interaction. Group A started working with the lure during Session 1, while Groups B and C experienced word conditioning during that session. As a result, statistical comparison of the training sessions examined the differences between Session 1 (Group A) and Session 2 (Group B and C), (labeled “Training 1”), between Session 2 (Group A) and Session 3 (Group B and C) (“Training 2”), and so on. Training Session 6 was not analyzed for Group A since the other groups did not have an equivalent sixth training session with the lure.

### 3.5. State Behaviors

Figure 5a–f show the percentage of time engaged in the six most common state behaviors: running, walking, staring and orienting towards the lure, being away from the lure, stalking, and engaging in alternative behaviors.

Running. There was a main effect of group, F (2, 14) = 37.04, *p* < 0.001, η_ρ_^2^ = 0.84, but not training session on percentage of time spent running, F (4, 56) = 0.468, *p* = 0.76 η_ρ_^2^ = 0.03. Pairwise t-test comparisons with Bonferroni corrections showed that Group B and Group C ran more than Group A, *p* = 0.001 and *p* = 0.02, respectively. This significant difference in percentage of time spent running for Group A can be viewed as an additional metric of the efficacy of the e-collar training as Group A stopped running once chasing was inhibited. Additionally, Group B ran significantly more than Group C, *p* < 0.001.

There was a significant interaction between training session and group, F (8, 56) = 4.48, *p* < 0.001, η_ρ_^2^ = 0.39. Simple effects pairwise t-test comparison analyses with Bonferroni corrections showed significant differences across the training sessions and groups. Group B spent significantly more time running than Group A across all matched training sessions, *p* < 0.05. Additionally, Group B spent significantly more time running than Group C in the first (*p* < 0.001), second (*p* < 0.001), and third sessions of training (*p* < 0.001), when the lure was stationary for Group C, but there were no significant differences in the remaining sessions, when the lure was moving for both groups. Group C ran significantly more than Group A in the fifth training session, *p* = 0.005. When analyzing within group differences, Group C spent significantly more time running in Training 5 than Training 2, *p* = 0.02, or Training 3, *p* = 0.004, when the lure was stationary.

Walking. There was no main effect of training session on the percentage of time spent walking, F (4, 56) = 0.77, *p* = 0.55, η_ρ_^2^ = 0.52, but the effect of group was significant, F (2, 14) = 17.58, *p* < 0.001, η_ρ_^2^ = 0.72. Pairwise comparisons with Bonferroni corrections showed that Group A walked significantly more than Group B, *p* < 0.001, and Group C, *p* = 0.02, while Group C walked significantly more than Group B, *p* < 0.001.

There was a marginally significant interaction between training session and group, F (8, 56) = 2.12, *p* = 0.048, η_ρ_^2^ = 0.23. Simple effects pairwise comparison analyses with Bonferroni corrections showed Group A spent significantly more time walking than Group B across all equivalent training sessions, *p* < 0.05. Group C spent significantly more time walking than Group C in the first, *p* = 0.004, and second training sessions, *p* = 0.03. Group A walked significantly more than Group C in Training 4, *p* = 0.004. There were no significant differences within groups for time spent walking across the training sessions.

Staring. Mauchly’s test indicated a violation of sphericity for the percentage of time staring, so Greenhouse–Geisser corrections were used. There was no main effect of training session on the percentage of time spent staring and orienting to the lure, F (1.73, 24.15) =2.15, *p* = 0.14, η_ρ_^2^ = 0.13, but there was a main effect of group, F (2, 14) = 6.06, *p* = 0.01, η_ρ_^2^ = 0.46. Pairwise t-test comparisons with Bonferroni corrections showed that Group A spent more time staring once chasing was inhibited compared to Group B, *p* = 0.04, and Group C, *p* = 0.02.

There was no significant interaction between training session and group, F (3.451, 24.15) = 0.69, *p* = 0.59, η_ρ_^2^ = 0.09.

Being away from the lure. There was no main effect of training session on percentage of time spent at the starting point, F (4, 56) = 0.17, *p* = 0.95, η_ρ_^2^ = 0.01, but there was a main effect of group, F (2, 14) = 6.61, *p* = 0.01, η_ρ_^2^ = 0.49. Pairwise *t*-test comparisons with Bonferroni corrections showed that Group A spent significantly more time away from the lure than Group B, *p* = 0.008.

There was a significant interaction between training session and group, F (8, 56) = 6.16, *p* < 0.001, η_ρ_^2^ = 0.49. Pairwise *t*-tests with Bonferroni corrections showed significant differences across the training sessions and groups. Group A spent significantly more time away from the lure than both Groups B and C for Training 1, *p* < 0.05. When the lure was stationary, Group C spent significantly more time away from it compared to Group B for Training 1, *p* < 0.001, and Training 2, *p* < 0.04.

There were significant differences between Group A and Groups B and C for the last two training sessions, *p* < 0.05. Group A spent significantly more time away from the lure in Training 4 and Training 5, *p* < 0.02, compared to Training 1. During Training 1, Group A dogs were away from the starting point, chasing the lure, but in the later sessions, they were inhibited in chasing and stayed at the starting point. When the lure was stationary, Group C spent more time away from it compared to Training 4, *p* < 0.001, and Training 5, *p* = 0.003, when it was finally moving.

Stalking. Mauchly’s test indicated that there was a violation of sphericity for the percentage of time spent stalking, so Greenhouse–Geisser corrections were used. There was no main effect of training session, F (1.37, 19.12) = 0.92, *p* = 0.38, η_ρ_^2^ = 0.06, or of group, F (2, 14) = 1.43, *p* = 0.27, η_ρ_^2^ = 0.17, on percentage of time spent staring and orienting to the lure. The interaction between training session and group was also not significant, F (2.73, 19.12) = 1.52, *p* = 0.24, η_ρ_^2^ = 0.18.

Engaging in Alternative Behavior. There was no significant main effect of training session, F (4, 56) = 0.58, *p* = 0.68, η_ρ_^2^ = 0.04, or group, F (2, 14) = 3.53, *p* = 0.06, η_ρ_^2^ = 0.34, on the percentage of time engaged in an alternative behavior. The interaction of these two factors was also not significant, F (8, 56) = 1.75, *p* = 0.11, η_ρ_^2^ = 0.20.

The point behaviors scratching, sniffing, shake-off, and yawning occurred infrequently and thus were not analyzed.

Overall, vocalizations were the most coded point behavior, and barking was the most common type of vocalization across the three groups (Figure 6). Shapiro–Wilk tests indicated significant deviations from normality, so non-parametric Kruskal–Wallis tests were conducted on vocalizations. The Kruskal–Wallis tests indicated statistically significant differences across the groups for yelping, H (2) = 7.14, *p* = 0.03. Post hoc analysis using Dunn’s procedure for pairwise comparisons with Bonferroni corrections showed that Group A yelped significantly more than Group B, *p* = 0.02.

There were no significant differences between groups for barking and whining: H (2) = 1.27, *p* = 0.53 and H (2) = 0.01, *p* = 0.99, respectively.

## 4. Discussion

This study compared the efficacy and welfare impacts of inhibiting chasing behavior with aversive and non-aversive methods. Under the conditions we implemented, chasing behavior was rapidly inhibited with e-collar stimulation whereas none of the dogs trained in either of the two conditions using non-aversive methods successfully inhibited chasing. We did not observe negative welfare impacts in the dogs trained with e-collars beyond presumably pain-induced yelps in immediate response to the electric shocks.

We designed our study’s methods around current trainer practices [24,25,26,27,32] as well as the recommendations of our senior trainer while attempting to keep continencies as simple as possible. Group A dogs received positive punishment (the addition of an aversive stimulus to reduce the likelihood a behavior recurs) via e-collar stimulation that was directly conditioned with contact with the moving lure. We also implemented the word “banana” immediately prior to shock administration as a warning signal, similar to the light offset in Solomon and Wynne [4]. E-collar training procedures often recommend determining the lowest level a dog will respond to and then increasing intensity gradually [24,25,33,34], and this method was used in the China et al. [11] and Cooper et al. studies [12]. However, we were mindful of the concern of stimulation habituation that occurs when animals are exposed to gradually increasing shock intensities [35,36] and were cognizant of the literature advocating for administering the most intense, punishing stimulus without causing damage [35,37,38]. Consequently, all the dogs in our study received the same starting intensity level that only increased or decreased based on the trainer’s perception of progress. Given that most dogs stopped running towards the lure after a small number of shocks, the functionality of the application of positive punishment in our study appears to be demonstrated.

Group B and C dogs were conditioned to the word “banana” by pairing it with the dropping of a treat in the bowl akin to an emergency recall protocol [28,33,39]. However, given that chasing is inherently reinforcing for dogs [26,40], Groups B and C experienced an alternative reinforcer that may have been more salient than the food reward. If we had turned off the lure when the dogs contacted it, this would have canceled the alternative reinforcer but would have constituted negative punishment, thus making it difficult to parse out the driving principle in any observed behavior change. For example, Marschark and Baenninger [40] reported that blocking access to a sheep (negative punishment) was sufficiently aversive to decrease dogs’ attempts to approach sheep.

Whereas Group B dogs were exposed to the lure at high speed in order to provide a close comparison to stimulus conditions in Group A, Group C dogs were initially exposed to the stationary lure, and its speed was increased as the dogs successfully responded to “banana” and returned to the starting point to obtain a treat. This procedure is more like commonly recommended non-aversive training practices in which stimulus levels are increased as the learners demonstrate proficiency at each level of stimulus exposure [27]. Group C dogs showed mixed responses to “banana”, with some initial successful inhibition, but they started running once the lure was at high speed. This initial partial success suggests that a gradual shaping procedure entirely reliant on non-aversive methods might be adequate to train inhibition of lure chasing if applied more gradually over a longer time course.

As running decreased in Group A, the percentage of time spent staring and orienting toward the lure increased. In a study of racing greyhounds, dogs fixated on the area where they last saw the lure when it was removed [41]. As the authors suggested, while the greyhounds were prevented from accessing the lure, the fact that they still focused on its direction and could hear it moving could result in frustration that may include increased arousal [41]. So, while the dogs in Group A were inhibited from directly chasing the lure, the increased percentage of time spent staring might suggest that they were at a heightened state of arousal and possible frustration.

Vocalizations, particularly yelping, are a common metric of pain and distress [6,8,12]. Unlike Cooper et al. [12] where yelping behavior was only observed in a few dogs, all dogs yelped on shock stimulation in our study with only scarce occurrences of yelping for a few dogs in the other groups. This is consistent with Schilder and van der Borg’s claim that e-collar-shocked dogs experience some level of pain [6]. However, it is difficult to know the long-lasting impacts of these experiences.

The typically fast-moving activity of the dogs in this study made it impossible to code more subtle behaviors that have been reported as indicative of stress in previous studies such as lip licking, ear position, or body posture [6,8,18]. Other stress-associated behaviors like yawning or shake-offs were included in our ethogram [12,18] but occurred very infrequently.

The range of values of fecal cortisol in our sample is comparable to previously published values [42,43,44,45,46,47]. Because not all dogs provided a sample on all days, the sample size available here is too small to allow significant conclusions. Furthermore, the use of cortisol levels as stress indicators has been questioned because of the range of external variables that can confound results [48]. Our results are consistent with previous studies of e-collar use which have also reported mixed results [8,12,16,49].

### 4.1. Limitations

There are several limitations to this study, including a sample size limited by the availability of people who, though willing to have their dogs trained with e-collars, had not already implemented training of this type. It was also challenging to record subtle behaviors when the dogs were in motion in a large outdoor arena. An additional video analysis of behaviors with a more extensive ethogram and assessment outside of the training scenario, such as obedience work [6] or play with their owner [19], may have provided more insight into the potential stress dogs experienced. One design flaw that could be avoided in future studies was that dogs in Group A had one more session of training than those in Groups B and C (because Group B and C dogs had a conditioning session for Training 1, rather than training with the lure). This may have placed Group A dogs at an advantage when they entered the tests but, given the magnitude of the differences in the performance of the three groups at the end of Training Session 5, this seems unlikely.

Groups B and C might have learned more effectively if training had progressed slower with more conditioning sessions of the word “banana” with treats as well as more gradually shaped exposure to the increasing lure speed. Indeed, in a more ideal set-up, conditioning sessions would have lasted over several days, possibly with a longer delay between each vocalization of “banana” and the presentation of a food. A more complete design would have included tests of the effectiveness of the food reward and whether the dogs indeed learned the paired association between saying “banana” and delivering food. Group C provided some suggestion that the dogs had learned the association between the food reward and the spoken “banana”, but this was still quite minimal compared to an ideal scenario. However, for a fair comparison of the training techniques, we needed a protocol with a time frame matched across groups. Indeed, one argument in favor of training with e-collars is that the results are much faster than with alternative non-aversive methods [50,51].

It is possible that our choice of food as reinforcer was inappropriate. We chose this method due to its prevalence within the non-aversive dog training community as a means to teach an emergency recall [28,39]. However, given that the dogs we tested and trained were highly motivated to chase the lure as demonstrated in Day 1 exposure, a reinforcer that related more closely to chasing behavior, such as an opportunity to chase a toy like a ball or play with a flirt pole, might have been more successful [52]. Feuerbacher and Wynne [53] reported that food is typically the most effective reinforcer for dogs, but whether alternative forms of reinforcement would be more successful in certain contexts is not well studied. A recent study attempted to determine whether targeted play that replicated a step in a dog’s normal predatory sequence would increase dogs’ gaze and focus on their owners and provide an alternative way to stop chasing rather than suppression or environmental management [54]. Owners were provided instructions and were told to mimic the prey sequence with a toy, to simply toss a toy for their dog to chase, or not to engage with their dog at all (control). However, this study found no differences in gaze towards owners across any of the groups [54].

We also did not test whether the food rewards we deployed were highly valued. Following common practice in dog training, we asked owners to provide their dog’s highest-value treats, but it would have been interesting, had time permitted, to have carried out a preference assessment with a range of food and, possibly, toy rewards presented to the dog in the training context.

Our study did not investigate the longer-term impacts of the aversive training methods deployed. While some previous studies that have observed whether a dog was still likely to chase a stimulus after being shocked [5,8,10], the data on long-term ramifications are limited. Christiansen and colleagues assessed long-term behavioral impacts of aversive stimulation a year after training from owner reports [5], and Schilder and van der Borg attempted to assess long-term impacts on a dog’s behavior, but the time frame between receiving shocks during training and observing the later behavior was still relatively short [6]. More research is warranted to determine the welfare impacts of prolonged exposure to aversive stimuli.

### 4.2. Ethical Considerations

The decision to use or forgo aversive stimulation is one of the most fraught in contemporary discussions of dog training. Many authors argue that, given the findings of stress-related behavior, poorer obedience, and possible aggression, the risk is too great for aversive methods to be recommended [11,12,16,19,55,56]. However, painful experiences during training are not the worst possible consequences for a dog with behavior incompatible with human cohabitation. Problem behavior is one of the major causes of the relinquishment of family dogs to animal shelters [57,58], and chasing behavior can have deadly consequences. Dogs that chase livestock may be shot or otherwise euthanized, and dogs that chase cars may be killed in traffic. Data on the rates of these outcomes are unavailable, but they are not trivial considerations [9,10,13].

Lindsay attempted to formalize the decision when and whether to use aversive stimulation with the principle of Least Intrusive, Minimally Aversive (LIMA) dog training where trainers should first attempt to modify behavior using non-aversive methods and, only if those are ineffective, escalate to more aversive methods [59]. Implementation of the LIMA principle relies on a trainer having the ability to recognize that a less aversive method has been implemented correctly and proven inadequate as well as the time to explore the impacts of different training methods. In an attempt to extend the LIMA principle and reduce the likelihood of trainers using aversive methods, Fernandez proposed the Least Inhibitive, Functionally Effective (LIFE) model [60]. Fernandez argued that ethical training should offer animals a range of choices, recognize the underlying function of the problem behavior, and assess the success of a method, not just in its ability to modify the behavior, but in its ability to meet the function of the original behavior.

The practical applicability of either the LIMA or LIFE models is likely limited, at least in the United States where dog trainers are not required to be licensed or certified. In a sample of highly rated dog trainers, more than half did not document any relevant education [61]. Additionally, most dog owners train their pets themselves rather than seeking assistance from professional dog trainers [62,63], and e-collars are widely available for purchase [56]. The global market for e-collars was estimated in 2023 at $100 million with a 15% annual growth rate [64], indicating widespread public interest in this approach to controlling dog behavior. Aversive training methods may be used without any knowledge or awareness of their potential impacts [65]. This complex situation with important real-world impacts underscores the importance of disinterested empirical studies of the immediate and longer-term outcomes of different methods of dog training.

The speed and effectiveness with which the e-collar inhibited the dogs in this study from chasing may justify the limited number of painful stimulations the dogs experienced if the object the dogs were chasing had been something that could directly or indirectly cause them serious injury or death. Furthermore, the fact that this could be achieved within a single day would likely make this an approach an appealing one to many dog owners in addressing problem behavior. However, the success of expert trainers in a controlled environment cannot be assumed to be representative of the outcomes that would be achieved by lay people acting without much or any knowledge of animal behavior. Consequently, neither this study nor others in the current literature should be viewed as guidance for people seeking to modify their dog’s behavior.

## 5. Conclusions

In this study, we found that dogs trained by professional expert trainers with aversive e-collar stimulation successfully inhibited chasing a lure in a five-day study of training. Two groups of dogs trained with non-aversive methods were considerably less successful in the same brief time frame. These results suggest that e-collar training within strict protocols may be effective with minimal negative side effects, but it should be acknowledged that most training is not carried out under these ideal conditions.

## Figures and Tables

**Figure 1 animals-14-02632-f001:**
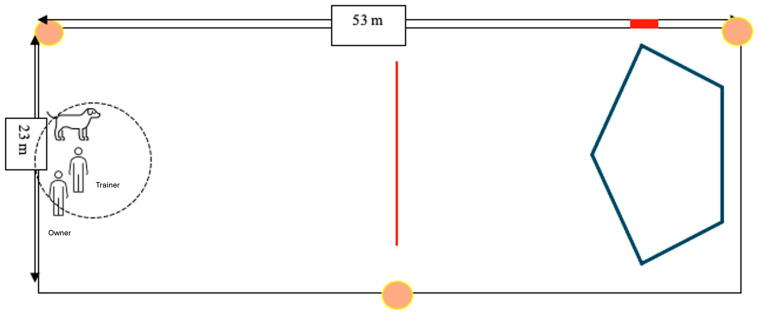
A schematic of the training arena with threshold indicators. The blue polygon indicates the location of the chase lure with the red rectangle indicating the location of the lure control box. The lure course is 45 m (approx. 150 ft) from the starting point. The red line indicates the testing threshold line, 23 m (approx. 75 ft) from the start position. The threshold line is 17 m (approx. 55 ft) from the closest part of the lure. The orange circles indicate the location of the cameras that recorded all the sessions. At the start of each session, the dog, owner, and trainer stood 45 m away from the lure course.

**Figure 2 animals-14-02632-f002:**
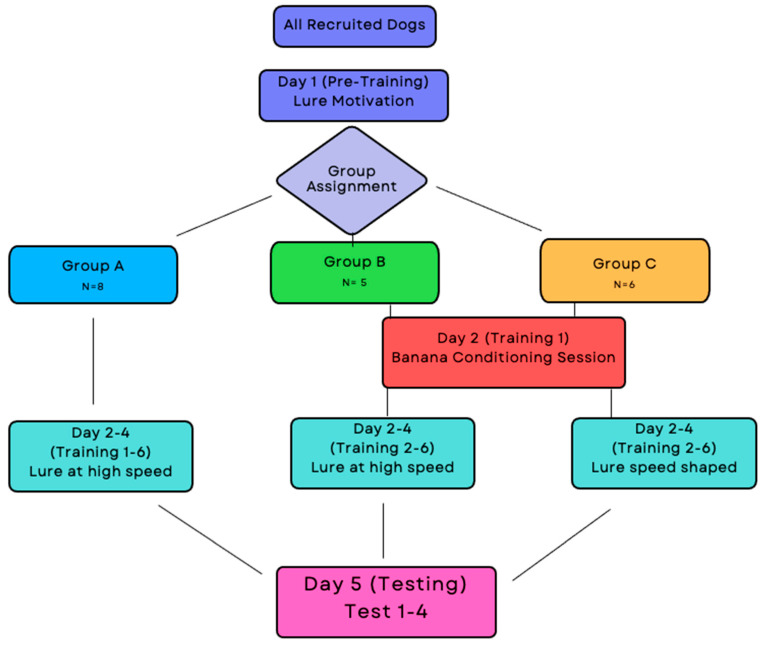
Flowchart for process of recruitment and procedure steps.

**Figure 3 animals-14-02632-f003:**
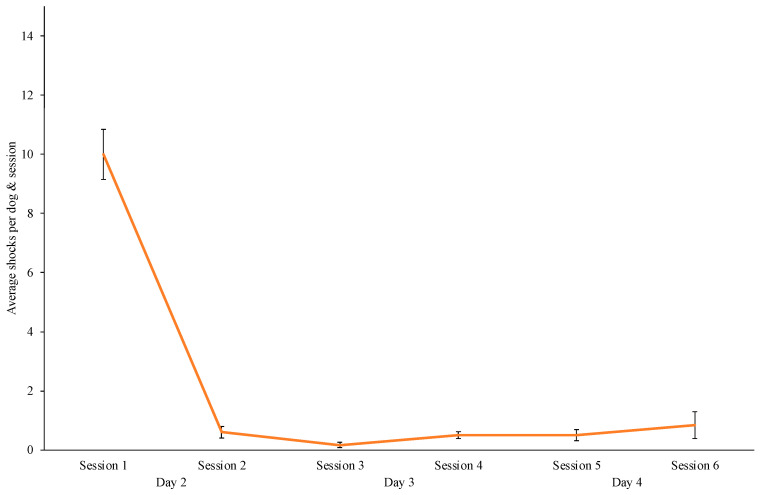
The average number of shocks per dog and training session for Group A. Note: Error bars represent standard errors.

**Figure 4 animals-14-02632-f004:**
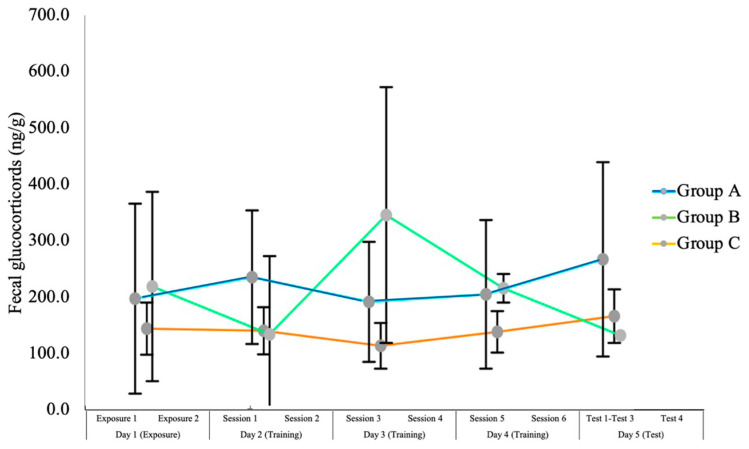
Mean fecal cortisol levels for each group of dogs on each day of this study. Note: Error bars indicate standard errors.

**Figure 5 animals-14-02632-f005:**
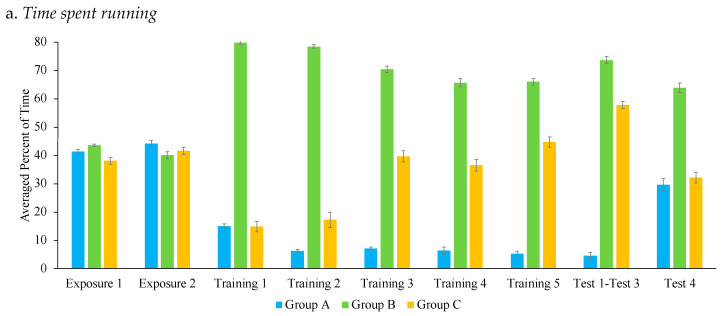
Comparison of average percentage of time in a state behavior across the three groups. Note: Error bars indicate standard errors. Sessions were matched for their equivalent exposure to the lure. For example, “Training 1” refers to Session 1 for Group A but Session 2 for Groups B and C.

**Figure 6 animals-14-02632-f006:**
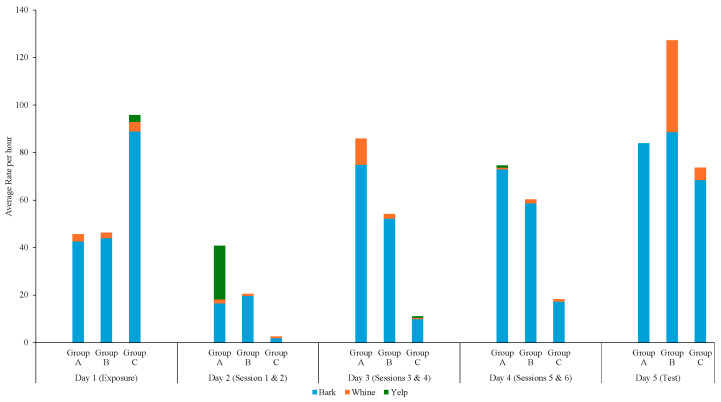
Average rate of vocalization per hour across the three groups.

**Table 1 animals-14-02632-t001:** Ethogram of behaviors coded in the video behavior analysis. Section A shows all the coded behaviors for the training sessions; Section B shows the alternative ethogram for Group B and Group C’s conditioning session (Day 2, Training 1).

Section A:		
Coded Behavior	Operational Definition	Behavior Type
Start position (Away)	Dog is at starting point of training arena; could be standing or laying down in the vicinity of the umbrella.	State
Off-Screen	Dog is not present on screen.	State
Stare	Dog is still, orientating to direction of the lure course. Could be laying down or standing.	State
Bark	Short, low deep vocalization.	Point
Yelp	Quick, sharp vocalization; may be once or over a few seconds.	Point
Whine	High-pitched vocalization, may be quick or over a prolonged period of time.	Point
Growl	Low-pitched, deep rumbling vocalization.	Point
Stalk	Dog will be standing upright, still, near lure, or in “play bow” position as if “hunting” lure.	State
Running	Dog is moving at a fast pace; at some point, all four paws will be off the ground as it moves.	State
Walking	Dog is moving at a slow to moderate pace; movement will vary between 2 and 3 paws on the ground at a time.	State
Yawning	Dog opens mouth wide, may occur with or without vocalization.	Point
Shake-off	Dog rapidly moves body and/or head, like how a dog may shake off water after a bath.	Point
Scratching	Dog stops whatever it is currently doing to use one limb to make repeated contact with its back or neck.	Point
Sniffing	Dog directs nose downward or upward to explore an item or substrate for longer than 1 s, end of sniffing bout signified by dog lifting its head which can be accompanied by walking away from the original focal object.	Point
Alternative Behavior	Dog is engaged in an activity that is not chasing, running, or standing at the start point. Behaviors can include attempting to elicit play from the humans in the training arena, finding an object to interact with, or rolling on the ground.	State
Section B: Section A state behaviors, bark, yelp, whine, growl, yawning, shake-off, scratching, and sniffing were included as part of the coding scheme for Group B and Group C’s Day 2, Training 1 conditioning sessions. Additional coded behaviors are listed and defined below.		
**Coded Behavior**	**Operational Definition**	**Type of Code**
Escape	Dog is attempting to exit penned-in area to get to open lure arena; behaviors can include pawing at pen and jumping.	State
Eat	Dog is ingesting presented food reward.	Point
Sit	Dog has forelimbs extended and is resting on bent hind limbs; can co-occur with “eat”.	Point
Lay	Dog is prostrate on the ground, forelimbs may be tucked under body or extended flat on ground in front of body; can co-occur with “eat”	Point
Offer	Dog presents a behavior, like sit, paw, or lay down, when treat not present but anticipating treat reward	Point

**Table 2 animals-14-02632-t002:** Summary of participating dogs.

Dog Name	Age (in Months) at Time of Study	Breed	Assigned Group
Calypso	30	German Shepherd	Group B
Chief	18	Belgian Malinois	Group A
Gizmo	18	Pitbull	Group C
Goose	9	Doberman	Group A
Hazel	24	German Shepherd	Group C
Jaxson	36	Border Collie	Group C
Loki	24	German Shepherd	Group A
Lola	10	Doberman	Group B
Major	24	Pitbull	Group C
Marley	18	German Shorthaired Pointer	Group B
Maya	30	Labrador Mix	Group B
Mochi	60	Alaskan Malamute	Group C
Mystery	24	German Shepherd	Group C
Rocky	14	Belgian Malinois	Group A
Ruby	10	English Labrador	Group A
Sage	20	Doberman	Group A
Tony	8	English Labrador	Group B

## Data Availability

The data presented in this study are available on request from the corresponding author.

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
