# Peer review of "Comparison of the Efficacy and Welfare of Different Training Methods in Stopping Chasing Behavior in Dogs"

_animals, 2024, doi:10.3390/ani14182632_

Round 1

Reviewer 1 Report

Comments and Suggestions for Authors

Comments in the attached pdf document.

Author Response

Overall, this is an excellent piece of research that speaks directly to some of the misinformation surrounding the efficacy of dog training methods.

--Thank you for these supportive comments. They are much appreciated.

I am concerned that the training procedures applied differed across the groups. For Group A, the association of "banana" and shock was established in the context of chasing the lure. For Groups B and C, the association of "banana" and food was established separately. I would have preferred that Group A experienced initial training of “banana” and shock separate from chasing the lure. However, this difference mimics how training would likely be conducted in the real world. I expect the biggest criticisms from the training industry will be that the word-food pairings were minimal and there needed to be an attempt to quantify how well the Groups B and C dogs learned the association. There was also no mention of food deprivation or consideration of each dog's motivation for the type of treat offered. Maybe these concerns are addressed in analyzing the mean rate per hour of eating treats, but I’m unclear on how to interpret that measure. Also, the time between “banana” and food repetitions was very short (5 secs). Lengthening that interval would be predicted to lead to stronger conditioning (the association between the word and the food would be more salient).—add in some type of wording   

We have added additional information in the limitations section about the lack of tests for whether the association of word with food was learned as well as the issue of type of reinforcer. We expand on how a study could be strengthened if timeframe were not a concern.

In my view, the most significant weakness of the research was the attempt to assess welfare. Based on the results, there is no doubt that the e-collar procedure was overwhelmingly more effective than the reward-based procedure. However, I felt that the question of the dogs' welfare was inadequately assessed. Several behavioral indications of stress occurred at such a low frequency that they could not be analyzed statistically. Perhaps with a more extensive ethogram, differences might have emerged or, if not, it would have been more convincing to conclude that e-collars do not significantly compromise welfare. And there were only a few fecal samples to be analyzed for cortisol levels so that wasn’t terribly informative.

The primary constraint on behavioral observation was not the level of detail in the ethogram but the fact that most dogs were in almost-continuous and fairly rapid motion in a large open field environment. That and the fact that the dogs did not show any overt welfare-impacted behavior. We have added a note suggesting extensive ethogram in conjunction our original note that some type of assessment outside of the direct training scenario might have provided greater insight.

 I noted that in Figure 5, the colors of the lines for Groups A and B appear to be the same.  

Figure 5 (Now Figure 4 given update from Reviewer 2) Group A bars to a darker blue for better differentiation between Group A and B on the chart.

A few minor things:  

  1. The authors say that the dogs had, at most, very limited experience with electronic collars. It would be helpful to explain what exactly “very limited” means.

Additional clarification added: (e.g.: owners may have introduced a collar once or twice to the dog but did not continue with e-collar training).

  1. The text says the dogs ranged in age from 9 mo to 6 yrs but according to the table, they ranged from 8 mo to 5 yrs. -

Confirmed in our data set that dog age ranged from 8 months to 5 years.

  1. For Group C, the authors report that once the dogs responded successfully to “banana” by returning to the bowl to eat the treat when the lure was not moving, they increased the speed to medium. I assume they skipped low because the dogs chasing the lure would have caught it almost immediately.

Correct, we added in a clarification to connect to the explanation of moving to high speed as well that dogs were catching and destroying the lure.

  1. From the Abstract: “In the 23 current study, dogs were trained across six sessions to desist from chasing a fast-moving lure in one 24 of three randomly-assigned conditions and then tested for retention and generalization in four further test trials.” The authors might want to revise this sentence to say “we attempted to train dogs to desist from chasing …”

Text updated as recommended: “we attempted to train”

Reviewer 2 Report

Comments and Suggestions for Authors

Thank you for the opportunity to review this paper. The research described in the article is highly important. Research of this type is very difficult to conduct and the care with which this study was conducted means that these results are extremely valuable.

I support publication of this work, but have a few questions and comments on the manuscript that the authors might consider in a revision.

Did the food treats function as reinforcers for Group B and C dogs? And, by association, did the word ‘banana’ become a conditioned reinforcer? It would be good to have some comment on this – it is mentioned, for example, that the level of the shock was adjusted over trials if it was observed that the dogs did not respond appropriately. This tells me that the researchers took care to ensure that the punishing consequence was indeed punishing. But it is difficult to judge whether the reinforcement used for the other two groups was an appropriate consequence, in that the treat was actually reinforcing to the dogs. The authors do address the question in the discussion where they debate the appropriateness of the reinforcer but it would strengthen the study if they had some data to show that the treats were reinforcing, even if that was just that the dogs returned and ate the treat when it was offered in the training phase.

I wonder if Figure 4 is really necessary? The results displayed here are adequately summarised in the text.

The results of Test 4 could receive more attention in the text. Although not all Group A dogs refrained from chasing the lure in this condition, the success rate here is quite astonishing given that no attempt was made during training sessions to generalise the response to other situations. I feel that the success – although not total – demonstrates a great deal of support for the e-collar condition. I note that in other studies mentioned by the authors, such as that by Dale et al. (2017), the effect of e-collar shock on dogs’ approach of animals (in this case, stuffed kiwi), no generalisation tests were performed and therefore it is difficult to say how effective such procedures are in ‘real’ situations (e.g., in the presence of a live kiwi). The fact that the authors here conducted a generalisation test, albeit not with a live animal, and that it demonstrated a relatively high level of success, is a very important point that should receive more attention. Of course, the ultimate generalisation test would involve a real prey animal but there are obvious ethical considerations which could prevent that type of test.

In the results section, it would be helpful to describe in the text the direction of the differences between groups. At present, the authors state that there were significant differences, but the direction of the differences is not stated and can only be extrapolated from the figures. I would also consider condensing the information in this section as it is rather repetitive to read. Perhaps the statistical information can be presented in a table?

The authors mention the small sample sizes as a limitation in the discussion section. I would encourage them to consider using behaviour analytic single-subject designs in their future research which can demonstrate clear effects of the independent variable without requiring large numbers of subjects. In the case of their present study, however, the results are very clear and I doubt that they would have been any different with a larger sample size. The issue that might be addressed here, however, is the two dogs that were excluded from Group A because they received a lot of shocks. What exactly happened with these two dogs? If the tests had included these dogs, might the results for Group A have showed that the e-collar was less effective than the present results do?

Comments on the Quality of English Language

The manuscript would benefit from a thorough proof-reading but the english language itself is fine

Author Response

Thank you for the opportunity to review this paper. The research described in the article is highly important. Research of this type is very difficult to conduct and the care with which this study was conducted means that these results are extremely valuable.

I support publication of this work, but have a few questions and comments on the manuscript that the authors might consider in a revision.

Thank you for these kind supportive comments. It means a lot to have our work recognized.

Did the food treats function as reinforcers for Group B and C dogs? And, by association, did the word ‘banana’ become a conditioned reinforcer? It would be good to have some comment on this – it is mentioned, for example, that the level of the shock was adjusted over trials if it was observed that the dogs did not respond appropriately. This tells me that the researchers took care to ensure that the punishing consequence was indeed punishing. But it is difficult to judge whether the reinforcement used for the other two groups was an appropriate consequence, in that the treat was actually reinforcing to the dogs. The authors do address the question in the discussion where they debate the appropriateness of the reinforcer but it would strengthen the study if they had some data to show that the treats were reinforcing, even if that was just that the dogs returned and ate the treat when it was offered in the training phase.

Unfortunately, we do not have data on the effectiveness of the food or the conditioned reinforcer for the food groups. We have added additional information in the  limitations section about our lack of testing for learning of the word-food association as well as the issue of type of reinforcer and how a study could be strengthened, if timeframe across groups was not a concern, with a preference assessment and testing of the association between “banana” and the presented food reward.

I wonder if Figure 4 is really necessary? The results displayed here are adequately summarised in the text.

Thank you for this observation. You are correct that it is unnecessary given the accompanying text, so it has been removed.

The results of Test 4 could receive more attention in the text. Although not all Group A dogs refrained from chasing the lure in this condition, the success rate here is quite astonishing given that no attempt was made during training sessions to generalise the response to other situations. I feel that the success – although not total – demonstrates a great deal of support for the e-collar condition. I note that in other studies mentioned by the authors, such as that by Dale et al. (2017), the effect of e-collar shock on dogs’ approach of animals (in this case, stuffed kiwi), no generalisation tests were performed and therefore it is difficult to say how effective such procedures are in ‘real’ situations (e.g., in the presence of a live kiwi). The fact that the authors here conducted a generalisation test, albeit not with a live animal, and that it demonstrated a relatively high level of success, is a very important point that should receive more attention. Of course, the ultimate generalisation test would involve a real prey animal but there are obvious ethical considerations which could prevent that type of test.

Your observation about Test 4 was also interesting to us. We viewed this study as a proof of concept using an object that the dogs were willing to chase but where contact with the stimulus object could not harm either the dogs or the object chased (at least, not beyond our ability to repair it without ethical implications). Were we to move forward with future studies we would work with dogs that chase cars. This is a major cause of harm for and ethically easier to work with than live prey chasing, given that a dog couldn’t harm a car and we could control movement of the car to minimize risk to the dog.

In the results section, it would be helpful to describe in the text the direction of the differences between groups. At present, the authors state that there were significant differences, but the direction of the differences is not stated and can only be extrapolated from the figures.

Additional clarification of how groups differed has been added to the text.

 I would also consider condensing the information in this section as it is rather repetitive to read. Perhaps the statistical information can be presented in a table?

Thank you for the suggestion. We decided to maintain the information in the text because the different analyses belong in different parts of the narrative. If we put them all together in a table it would become difficult for a reader to navigate. 

The authors mention the small sample sizes as a limitation in the discussion section. I would encourage them to consider using behaviour analytic single-subject designs in their future research which can demonstrate clear effects of the independent variable without requiring large numbers of subjects. In the case of their present study, however, the results are very clear and I doubt that they would have been any different with a larger sample size. The issue that might be addressed here, however, is the two dogs that were excluded from Group A because they received a lot of shocks. What exactly happened with these two dogs? If the tests had included these dogs, might the results for Group A have showed that the e-collar was less effective than the present results do?

Thank you for the suggestion of single-subject BA designs. It’s certainly something to consider, for the future. As you note, a larger sample size would likely not impact the current study conclusions, and we added a note that the inclusion of the two excluded dogs would not impact the results of the study.

See Section 3.1

Inclusion of these dogs’ data would not have altered the overall conclusions of this study.